# The protein segregase VCP/p97 promotes host antifungal defense via regulation of SYK activation

Zhugui Shao[1,2,3], Li Wang[3,4], Limin Cao[3], Tian Chen[1,5], Xin-Ming Jia[4], Wanwei Sun[1,2]*, Chengjiang Gao [1,2]*, Hui Xiao[3]*

**1** Key Laboratory of Infection and Immunity of Shandong Province & Key Laboratory for Experimental Teratology of Ministry of Education, Shandong University, Jinan, Shandong, P. R. China, **2** Department of Immunology, School of Basic Medical Sciences, Shandong University, Jinan, Shandong, P. R. China, **3** Key Laboratory of Immune Response and Immunotherapy, Shanghai Institute of Immunity and Infection, University of Chinese Academy of Sciences, Chinese Academy of Sciences, Shanghai, P. R. China, **4** Department of Infection and Immunity, Clinical Medicine Scientific and Technical Innovation Center, Shanghai Tenth People's Hospital, School of Medicine, Tongji University, Shanghai, P. R. China, **5** Department of Pathogenic Biology, School of Basic Medical Sciences, Shandong University, Jinan, Shandong, P. R. China

\* sunwanwei2021@sdu.edu.cn (WS); cgao@sdu.edu.cn (CG); huixiao@siii.cas.cn (HX)

**Data Availability Statement:** All relevant data are within the manuscript and its Supplemental Information files.

## Abstract

C-type lectin receptors (CLRs) are essential to execute host defense against fungal infection. Nevertheless, a comprehensive understanding of the molecular underpinnings of CLR signaling remains a work in progress. Here, we searched for yet-to-be-identified tyrosine-phosphorylated proteins in Dectin-1 signaling and linked the stress-response protein valosin containing protein (VCP)/p97 to Dectin-1 signaling. Knockdown of VCP expression or chemical inhibition of VCP's segregase activity dampened Dectin-1-elicited SYK activation in BMDMs and BMDCs, leading to attenuated expression of proinflammatory cytokines/chemokines such as TNF-α, IL-6 and CXCL1. Biochemical analyses demonstrated that VCP and its cofactor UFD1 form a complex with SYK and its phosphatase SHP-1 following Dectin-1 ligation, and knockdown of VCP led to a more prominent SYK and SHP-1 association. Further, SHP-1 became polyubiquitinated upon Dectin-1 activation, and VCP or UFD1 overexpression accelerated SHP-1 degradation. Conceivably, VCP may promote Dectin-1 signaling by pulling the ubiquitinated SHP-1 out of the SYK complex for degradation. Finally, genetic ablation of VCP in the neutrophil and macrophage compartment rendered the mice highly susceptible to infection by *Candida albicans*, an observation also phenocopied by administering the VCP inhibitor. These results collectively demonstrate that VCP is a previously unappreciated signal transducer of the Dectin-1 pathway and a crucial component of antifungal defense, and suggest a new mechanism regulating SYK activation.

**Funding:** This study was supported by the National Key Research and Development Program of China (2021YFA1301400 to H.X.); National Natural Science Foundation of China grants (32030040, 81720108019 to H.X.); the Strategic Priority Research Program of the Chinese Academy of Sciences (XDB0940102 to H.X.). The funders had no role in study design, data collection and analysis, decision to publish, or preparation of the manuscript. The authors received no specific funding for this work.

**Competing interests:** The authors have declared that no competing interests exist.

## Author summary

C-type lectin receptors (CLRs) are crucial for defending against fungal infections, but the details of CLR signaling are still being explored. This study identified a new role for the stress-response protein VCP/p97 in Dectin-1 signaling, which is key for host defense. When VCP is reduced or its activity is blocked, Dectin-1's ability to activate SYK is weakened, leading to lower levels of inflammatory responses. Upon Dectin-1 activation, VCP and UFD1 integrate into a SYK-SHP-1 complex. Diminished VCP levels paradoxically augment the interaction between SYK and SHP-1. Evidence suggests that VCP facilitates Dectin-1 signaling by sequestering ubiquitinated SHP-1 from the SYK complex. Mice with a genetic deficiency of VCP in neutrophils and macrophages exhibit heightened susceptibility to *Candida albicans* infection, a phenotype recapitulated by pharmacological VCP inhibition. Collectively, these findings underscore VCP's indispensable role in the Dectin-1 signaling axis and underscore a novel regulatory mechanism in SYK activation.

## Introduction

Fungal infection poses a great threat to public health throughout the world due to its high mortality rates and increased costs of treatment [1]. The risk of infection is especially high among immunocompromised individuals, including patients under intensive care or chemotherapy [2]. *Candida albicans* is an opportunistic pathogen that causes local mucosal infections as well as life-threatening systemic infections in the bloodstream. Meanwhile, the treatment of *Candida albicans* infection remains challenging. Therefore, it is necessary to better understand the mechanistical interaction between host immune cells and fungal pathogens to uncover novel target therapy to combat *C. albicans* infection [3].

Host immune cells utilize germline-encoded receptors, collectively termed pattern recognition receptors (PRRs), to detect conserved microbial components, known as pathogen-associated molecular patterns (PAMPs) [4]. During the fungal infection CLRs include Dectin-1 (also known as CLEC7A), Dectin-2 and Mincle expressed on macrophages, dendritic cells or neutrophils play a pivotal role in the recognition of fungal cell wall components such as β-glucans, a-mannans, and glycolipids [5–7]. Dectin-1 is an archetypical CLR that detects β-glucans in fungal cell walls and triggers direct cellular antimicrobial activity. Dectin-1 signals through a 'hemITAM' motif, an immunoreceptor tyrosine-based activation motif containing a single 'YxxL', that becomes phosphorylated by SRC family kinases after receptor engagement [8]. This phosphorylation allows Dectin-1 to recruit the adaptor protein SHP2 and spleen tyrosine kinase SYK to form the receptor-proximal signalosome on the cell membrane [9]. Subsequently, activated SYK transduces the signal to phospholipase lipase Cγ2 (PLCγ2) [10], and PKCδ [11], which phosphorylates central adaptor caspase recruitment domain-containing protein 9 (CARD9) to activate downstream NF-κB and MAPKs [12–14]. These signaling pathways turn on a series of antifungal mechanisms, including proinflammatory cytokines and chemokines release, phagocytosis, and ROS production, ultimately leading to fungi clearance [15].

SYK functions as a signaling hub within the CLR signaling complex whose activity needs to be tightly controlled during fungal infections. In resting cells, SYK adopts a "closed" autoinhibitory structure. Upon ligand engagement, SYK switches to an "opened" conformation due to its binding to the upstream receptor or adaptor, permitting the phosphorylation by the upstream kinase SRC [16,17]. We previously found that the E3 ligase TRIM31 catalyzes polyubiquitination of SYK to promote its binding with Dectin-1, thus positively regulating SYK

activation [18]. On the other hand, SHP-1, an important phosphatase for SYK, interacts with SYK and counteracts its phosphorylation to negatively regulates the Dectin-1 signaling [19,20]. SYK signaling can also be terminated through SYK degradation, modulated by the E3 ubiquitin ligase Casitas B-lineage lymphoma (CBL) [21,22]. Therefore, complex positive and negative regulators are required to precisely control SYK activation.

VCP/p97 is an AAA+ ATPase protein that is conserved across diverse species [23]. The primary function of ATP-driven VCP/p97 is to assist protein degradation or recycling through the proteasome pathway and plays crucial roles in ubiquitin-dependent protein quality control and signaling events. VCP works with multiple cofactor proteins including UFD1 to extract diverse ubiquitin-labeled proteins from membranes or protein complexes in many cellular contexts [24]. Recent studies have revealed a critical role of VCP in the pathogenesis of Inclusion Body Myopathy disease, cardiovascular disease, Huntington's disease, *etc.*[25–27]. VCP is involved in diverse physiological processes to regulate protein degradation, such as autophagy, endoplasmic reticulum (ER)-stress, chromatin-associated processes and stress-granule disassembly [28–30]. Although VCP participates in a number of cellular activities based on its segregase activity, the role of VCP in regulating antifungal immunity is unexplored.

Here, we identified VCP as a new tyrosine phosphorylated protein upon Dectin-1 engagement, positively regulating antifungal immune responses upon *C. albicans* infection. Using pharmacological inhibitors and genetic silencing approaches, we found that inhibition of VCP activity dampened the phosphorylation of SYK. Further, VCP also promotes the dissociation of SHP-1 from the SYK signalosome complex, relieving the negative impacts of SHP1 on SYK phosphorylation. By using VCP-deficient mice with *Vcp* being knocked out specifically in macrophage and neutrophil, we found VCP deficiency impaired SYK-mediated signaling and inhibited the proinflammatory cytokine production in the mouse model of systemic fungal infection, rendering mice highly susceptible to fungal infection *in vivo*. Altogether, our results uncovered VCP as a novel regulator of SYK activation and therefore crucial component of antifungal innate immunity.

## Results

### VCP is tyrosine phosphorylated upon Dectin-1 signaling activation

Engaging Dectin-1 triggers the activation of tyrosine kinases SRC/LYN and SYK, thereby promoting the phosphorylation of a myriad of signaling molecules [31,32]. Bone marrow-derived DCs (BMDCs) stimulated with Zymosan depleted (ZymD), a specific ligand for Dectin-1, multitudes of bands of induced tyrosine phosphorylation modifications were observed in the vicinity of 70 to 130 kD, and prominent tyrosine-phosphorylation on proteins were expected with MWs of 70–85, 100 or 125 kDa (**Fig 1A**). Since SHP2 (~72 kDa), SYK (~72 kDa), PLCγ2 (~150 kDa), and PKC-δ (~78 kDa) have been reported to play important roles in antifungal immune response following respective tyrosine phosphorylation, we sought to identify other phosphorylated molecules involved in this process, focusing on proteins with molecular weights approximately 100 kDa. To this end, the corresponding SDS-PAGE gel containing proteins of M.W. 100 kDa was sliced and digested, followed by mass spectrometry analysis (**Fig 1B**). Interestingly, VCP was identified as one of the tyrosine phosphorylated proteins with relatively high Mascot score. In support of VCP's phosphorylation by Dectin-1 signaling, anti-p-Tyr immunoprecipitation pulled down more VCP proteins in BMDCs stimulated with ZymD (**Fig 1C**). Likewise, immunoprecipitated VCP probed with anti-p-Tyr showed that VCP was marginally phosphorylated in the steady state, but became prominently phosphorylated after ZymD stimulation (**Fig 1D**). Consistently, ZymD stimulation also increased the tyrosine phosphorylation on VCP in BMDMs, and inhibiting kinases SRC and SYK with PP2

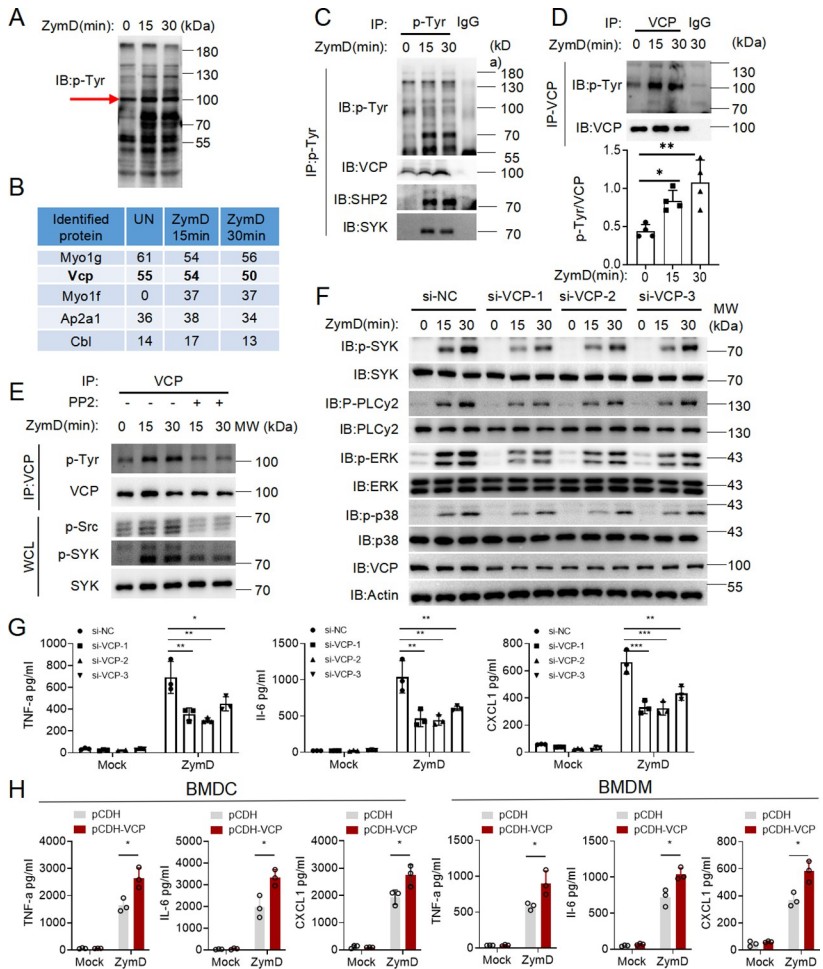

**Fig 1. VCP is a tyrosine phosphorylated protein promoting Dectin-1 signaling.** (A). Immunoblot analysis of proteins with phosphorylated tyrosine (p-Tyr), in lysates of WT BMDCs left untreated (0 min) or treated for 15 or 30 min with ZymD (100 μg/ml). The red arrow indicates the target bands. **(B)** About 100 kDa proteins identified by mass spectrometry are shown. **(C)** IP analysis of cell lysates from BMDCs treated with Dectin-1 ligand ZymD (100 μg/ml) for 15 and 30 minutes, and IP with p-tyrosine antibody or an IgG control. Precipitates immunoblotted with the VCP, SYK, SHP2 and phosphorylated tyrosine antibody. **(D)** IP analysis of cell lysates from BMDCs treated with Dectin-1 ligand ZymD (100 μg/ml) for 15 and 30 minutes, and IP with VCP-specific antibody or an IgG control. Precipitates probed for VCP and phosphorylation-tyrosine antibody. Densitometric quantification of p-Tyr/VCP as shown in Fig 1D from four independent experiments were measured by Image J. **(E)** Immunoassay of lysates of BMDMs pretreated for 1 h with vehicle (-), the Src inhibitor PP2 (5 μM) followed by treatment for 0, 15 or 30 min with ZymD (100 μg/ml). Assessed by immunoprecipitation with anti-VCP, followed by immunoblot analysis. **(F)** Western blot analysis of the phosphorylation of signaling proteins in BMDMs transfected with si-control (si-NC) and VCP knock down (si-VCPs) and then stimulated with ZymD for indicated times. **(G)** Enzyme-linked immunosorbent assay (ELISA) of cytokines and chemokines in supernatants of BMDMs derived from si-NC and si-VCPs group, left unstimulated (Mock) or stimulated with ZymD (100 μg/ml) for 24 h. **(H)** ELISA of cytokines and chemokines in pCDH-EV or pCDH-VCP lenti-virus infected BMDMs or BMDCs unstimulated (Mock) or stimulated with ZymD (100 μg/ml) for 24 h. Data are shown as the mean ± SD of three biological replicates and were analyzed using one-way ANOVA (G) and an unpaired two-tailed Student's t test (H). (*: p<0.05; **: p<0.01; ***: p<0.001; ****: p<0.0001).

nulled ZymD's such effect (**Fig 1E**). Moreover, VCP was also tyrosine phosphorylated in HEK293T cells co-transfected with SYK and Dectin-1 following ZymD stimulation (**S1A** and **S1B Fig**). Together, we identified VCP as one of the previously unrecognized tyrosine-phosphorylated proteins in Dectin-1 signaling.

## VCP promotes Dectin-1 signaling

Next, we employed siRNA knockdown to investigate the role of VCP in Dectin-1 signaling. Three independent siRNAs against VCP mRNA resulted in a modest but significant reduction in VCP protein (**Fig 1F**). Following ZymD stimulation, SYK phosphorylation was substantially decreased in VCP-knockdown cells as compared to the control siRNA-transfected WT cells (**Figs 1F** and **S2A**). Correspondingly, ZymD-induced tyrosine phosphorylation on PLC-γ2 was reduced in VCP-knockdown BMDMs (**Figs 1F** and **S2A**), indicative of impaired SYK activation. Also, the activation of downstream MAPKs, ERK and p38 were attenuated in VCP-knockdown cells (**Figs 1F** and **S2A**). Corroborating their defects in Dectin-1 signaling, VCP-knockdown cells produced much less proinflammatory molecules such as TNF-α, IL-6 and CXCL1 than the control cells (**Fig 1G**). To complement the siRNA knockdown strategy, we also constructed recombinant lentiviruses expressing FLAG-VCP and transfected them into BMDMs and BMDCs, respectively. Compared to their respective control cells transfected with the empty lentiviral vector pCDH, BMDMs- and BMDCs-transfected with pCDH-VCP showed higher expression of TNF-α, IL-6 and CXCL1 (**Fig 1H**). Taken together, these data indicate that VCP positively regulates Dectin-1-induced signaling cascade and proinflammatory response.

## VCP's ATPase activity plays an important role in SYK activation

VCP is an AAA$^+$ type ATPase whose activity can be inhibited by several inhibitors [33], we then interrogated the role of its ATPase activity in Dectin-1 signaling by pharmacologic approaches. Remarkably, ZymD-induced SYK phosphorylation was significantly impaired by all four inhibitors of VCP (**Fig 2A**), among them, NMS-873 showed the most prominent effect. We further confirmed that NMS-873 inhibited SYK activation in a dose-dependent manner, and was highly effective even at 1 μM (**Fig 2B**). Of note, ZymA, a fungal cell wall component capable of engaging both Dectin-1 and TLR2, elicited a more pronounced signaling response compared to ZymD solely engaging Dectin-1. Following stimulation with Zymosan (ZymA), PKC-δ and IKKα/β were robustly phosphorylated in BMDMs. NMS-873 almost abolished the activation of SYK, PKC-δ and IKKα/β at higher dose (10 μM). Also, NMS-873 effectively abrogated the phosphorylation and activation of MAPKs such as JNK, ERK and p38 by ZymA stimulation (**Figs 2B** and **S2B**). Hence, we chose NMS-873 to block VCP activity hereafter.

Further, concurrent treatment with NMS-873 also blunted ZymD- or ZymA-induced TNF-α, IL-6 and CXCL-1 production in BMDMs (**Fig 2C**). Similarly, much fewer TNF-α, IL-6 and CXCL-1 were detected in the supernatants of BMDCs stimulated by ZymD with NMS-873 (**Fig 2D**). Hence, inhibiting VCP with NMS-873 could robustly impair Dectin-1 signaling. We also employed another VCP inhibitor DBeQ to corroborate VCP's ATPase activity with SYK activation. Inhibiting VCP with DBeQ also robustly impaired Dectin-1 signaling (**S3A Fig**), leading to a reduction in ZymD- or ZymA-induced TNF-α, IL-6 and CXCL-1 production in BMDMs (**S3B Fig**). To further interrogate the role of VCP's ATPase in Dectin-1 signaling, we implemented a reconstitution system suitable for the study of Dectin-1 signaling in HEK293T cells. Consistent with our previous study, reconstituting HEK293T cells with exogenous Dectin-1 and SYK led to their responsiveness to ZymD stimulation. While co-expression of the wild-type VCP augmented SYK phosphorylation, concurrent expression of the ATPase-dead mutant of VCP (K542A) blunted SYK activation [34] (**Fig 2E**). These results collectively indicate VCP's ATPase activity plays an important role in the transduction of Dectin-1 signaling.

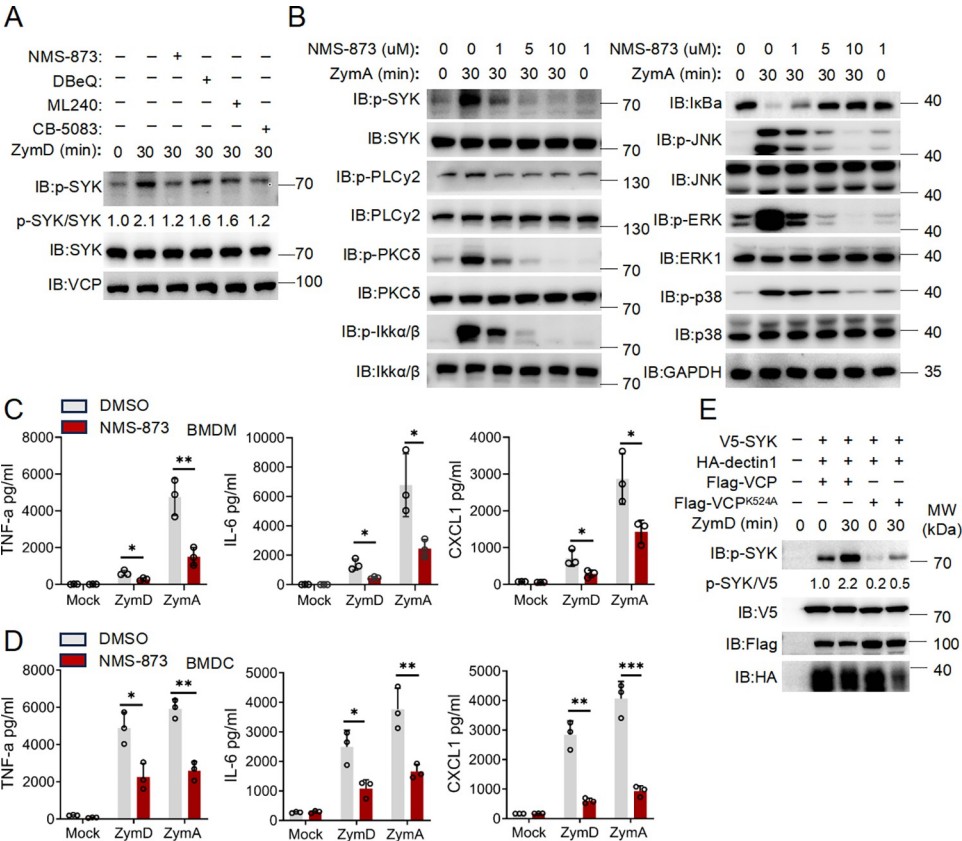

**Fig 2. VCP's ATPase activity is required for SYK activation. (A)** BMDMs were pretreated with vehicle (-) or the VCP inhibitor NMS-873, DBeQ, ML240, CB-5083 at 5 μM for 1 hour, followed by the cells were treated for either 0 or 30 minutes with ZymD (100 μg/ml) and then subjected to immunoblotting analysis using anti-p-SYK, anti-SYK, and anti-VCP antibodies. **(B)** Immunoblot analysis of whole-cell lysates of BMDMs derived from wild-type mice and left unstimulated (0 min) or stimulated for 30 min with ZymA (100 μg/ml) in the presence of different concentration of NMS-873 at 1, 5 and 10 μM were performed with indicated antibodies. **(C-D)** ELISA of TNF-a, IL-6 and CXCL1 in BMDMs or BMDCs unstimulated (Mock) or stimulated with either Zymosan depleted (ZymD) (100 μg/ml), zymosan-A (ZymA) (100 μg/ml) in the presence of DMSO or 0.5 μM of NMS-873 for 24 hours. **(E)** Immunoblot analysis of proteins from lysates of HEK293 cells transiently transfected with various combinations of plasmids expressing Flag-tagged wild-type VCP or mutant Flag-VCP$^{K524A}$ along with plasmid expressing HA-Dectin-1, V5-SYK. Then 36 hours later, the cells were left unstimulated or stimulated for 30 min with ZymD (100 μg/ml) before being immunoblotted with the indicated Abs. Data are shown as mean ± SD of three biological replicates and were analyzed using an unpaired two-tailed Student's t test (C-D). *$p < 0.05$, **$p < 0.01$, ***$p < 0.001$.

## VCP promotes the dissociation of tyrosine phosphatase SHP-1 from the SYK signalosome

In order to explore the underlined mechanism by which VCP regulates Dectin-1 signaling, we investigated what signaling molecules VCP may interact with upon Dectin-1 activation. By co-immunoprecipitation, we found that ZymD stimulation potentiated the interaction of VCP with SYK, and Dectin-1 in BMDMs, indicating that VCP is recruited to the SYK signalosome (**Fig 3A**). This observation was confirmed by the reconstituting Dectin-1-SYK system in HEK293T cells (**Fig 3B**). Co-transfections of plasmid encoding Flag-tagged VCP along with V5-SYK- or/and HA-Dectin-1-expressing plasmids to HEK293T cells for 48 hours were followed with ZymD stimulation for 15 min. Co-immunoprecipitation assay revealed that Flag-VCP associated with V5-SYK and HA-Dectin-1 regardless of ZymD stimulation (**Fig 3B**). VCP interacts with its cognate substrates through various adaptor molecules, we found its

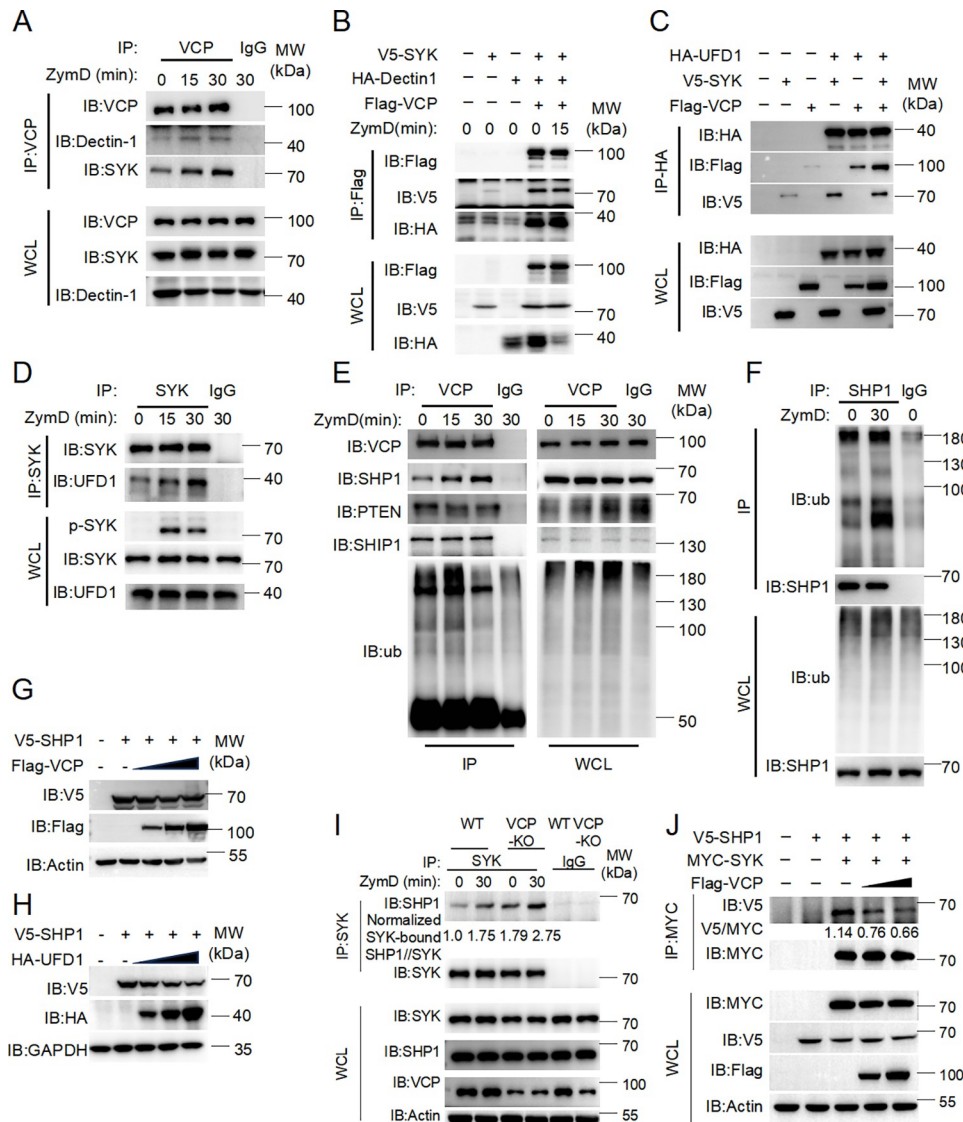

**Fig 3. VCP promotes the dissociation of tyrosine phosphatase SHP-1 from the SYK signalosome.** (A). Wild-type BMDMs were stimulated with ZymD for the indicated times, after which the cell lysates were subjected to immunoprecipitation (IP) with anti-VCP antibody, followed by Western blot analysis with anti-Dectin-1 and anti-SYK antibodies. **(B)** HEK293T cells were transfected with the indicated plasmids for 36 h. The cell lysates were subjected to IP with FLAG-M2 beads and then immunoblotted with anti-V5 and anti-HA, sequentially. **(C)** HEK293T cells were transfected with the indicated plasmids for 36 h. The cell lysates were subjected to IP with anti-HA beads and then immunoblotted with anti-V5 and anti-Flag antibodies, sequentially. **(D)** WT BMDMs were stimulated with ZymD for indicated times and the cell lysates were subjected to IP with anti-SYK, followed by Western blot analysis with anti-UFD1. **(E)** WT BMDMs were stimulated with ZymD for indicated times and the cell lysates were subjected to IP with anti-VCP, followed by Western blot analyses with anti-SHP1, anti-PTEN, anti-SHIP1, and anti-ub, respectively. **(F)** WT BMDMs were stimulated with ZymD for indicated times and the cell lysates were subjected to IP with anti-SHP-1, followed by Western blotting with anti-ub. **(G)** HEK293T cells were transfected with the indicated plasmids for 36 h. The cell lysates immunoblotted with anti-V5 and anti-Flag. **(H)** HEK293T cells were transfected with the indicated plasmids for 36 h. The cell lysates immunoblotted with anti-V5 and anti-HA. **(I)** WT and VCP deficient BMDMs were stimulated with ZymD for indicated times and cell lysates were subjected to IP with anti-SYK Ab, followed by Western blot analysis with anti-SHP1 Ab. **(J)** HEK293T cells were transfected with Myc-SYK, V5-SHP-1 and Flag-VCP. Followed by IP with anti-MYC, probed with indicated antibodies (left margins). Data are from one experiment representative of three.

adaptor UFD1 also interact with SYK (**Fig 3C**), and their interaction was enhanced by ZymD stimulation in BMDMs (**Fig 3D**). These results suggest that VCP and its accessary partner UFD1 may be recruited to the SYK signalosome upon Dectin-1 activation.

Interestingly, VCP also became more abundantly associated with tyrosine phosphatase SHP-1 after ZymD stimulation, although its interactions with other negative regulators such as PTEN and SHIP1 remained unaltered (**Fig 3E**). Furthermore, the interaction between VCP and SHP-1 was confirmed by their co-immunoprecipitation from HEK293T cells co-transfected with VCP- and SHP-1-encoding plasmids (**S4A Fig**). It is worth noting that SHP-1 became heavily polyubiquitinated in BMDMs treated with ZymD (**Fig 3F**). Considering the ternary complex containing VCP, UFD1 and NPLOC4 binds ubiquitinated proteins and is necessary for the export of misfolded proteins from the ER to the cytoplasm, where they are degraded by the proteasome, we sought to test whether VCP and UFD1 could act as the segregase recognizing the ubiquitinated proteins and pulling them out of the signalosome for degradation. Overexpression of SHP-1 along with SYK and Dectin-1 in HEK293T cells led to completely dephosphorylation of SYK (**S4B Fig**). Overexpression of VCP or UFD1 in HEK293T cells accelerated the degradation of SHP-1 in a dose-dependent manner (**Fig 3G** and **3H**). Upon co-expression of SHP-1 and VCP, treatment with the protein synthesis inhibitor cycloheximide (CHX) for different times led to an accelerated degradation of SHP-1 protein (**S4C Fig**). To determine how VCP mediates the degradation of SHP-1, we tested various inhibitors commonly used to study protein degradation, and found that VCP-induced SHP-1 degradation could be blocked by proteasome inhibitor MG132, but not by autophagy inhibitor Chloroquine (CQ). These results indicate that VCP may promote SHP-1 degradation through the proteasome (**S4D Fig**).

Further, we generated $Vcp^{fl/fl}$ mice and bred them onto Lyz2-Cre transgenic mice, creating a $Vcp^{fl/fl}$ Lyz2-Cre strain deficient in VCP in the neutrophil and macrophage compartment (hereafter referred to as VCP-cKO). We prepared BMDMs from $Vcp^{fl/fl}$ and VCP-cKO mice, followed by stimulation with ZymD. We found that SHP-1 was also recruited to the SYK complex following ZymD stimulation, and knockout of VCP led to much stronger SHP-1 association with SYK in BMDMs (**Figs 3I** and **S5A**). Overexpression of VCP attenuated the interaction between SYK and SHP-1 (**Figs 3J** and **S5B**). Next, to test whether VCP specifically targets SHP-1 degradation to regulate dectin-1 signaling, we used siRNA knockdown SHP-1 or TPI-1 to block SHP-1 activity. Compared with si-NC or DMSO group, si-Shp-1 or TPI-1 treatment enhanced the phosphorylation of Syk in $Vcp^{fl/fl}$ cells. However, knockdown of SHP-1 expression or blocking SHP-1 activity (TPI-1) led to comparable levels of ZymD-induced Syk phosphorylation between the WT and $Vcp$-deficient groups (**S5C** and **S5D Fig**). These results implicate that VCP negatively regulates the retention of SHP-1 in the SYK signalosomes during Dectin-1 signaling.

## VCP-deficiency disrupts Dectin-1 signaling to heat-inactivated *Candida albicans* in macrophages

We prepared BMDMs from $Vcp^{fl/fl}$ and VCP-cKO mice, followed by stimulation with heat-killed yeast-form of *C. albicans* (HKCA-Y). Immunoblotting assay demonstrated that HKCA-induced robust phosphorylation of SYK, PLC-γ2 and PKC-δ in WT BMDMs. However, the phosphorylation of SYK, PLCγ-2 and PKC-δ were attenuated in VCP-deficient BMDMs (**Figs 4A** and **S6A**). Similarly, the activation of ERK and p38 were also significantly impaired by VCP deficiency (**Figs 4A** and **S6A**). Also, ZymD-induced Dectin-1 signaling events, including the activation of SYK, PLC-γ2, PKC-δ, ERK and p38, were diminished in VCP-deficient

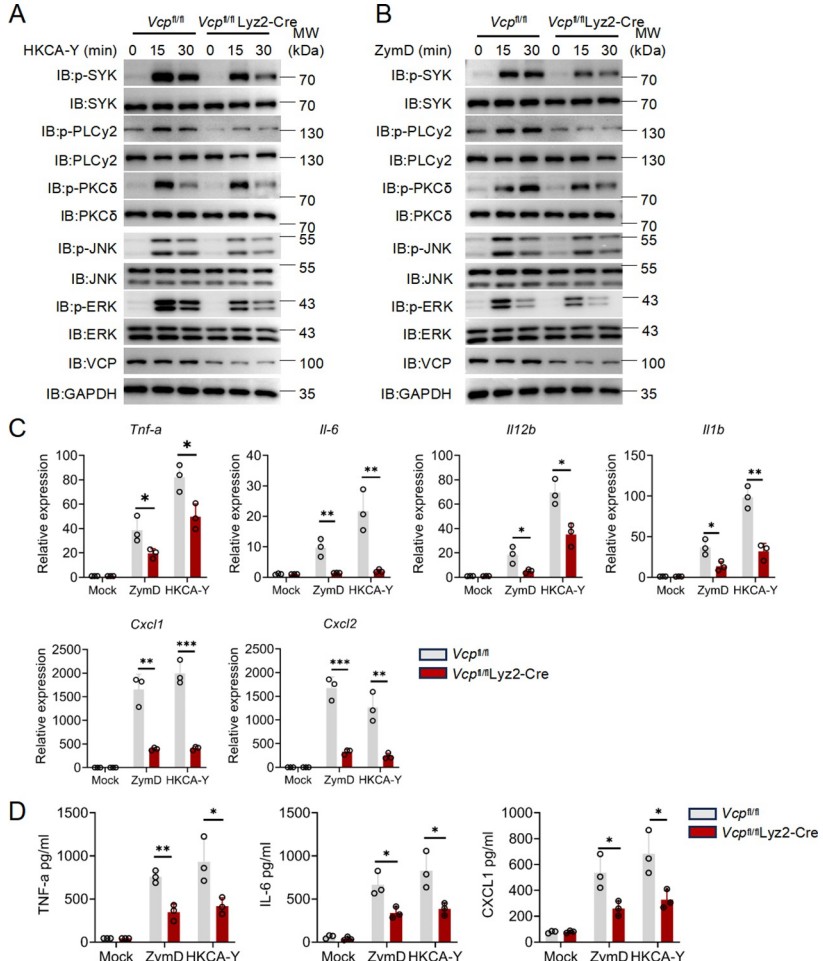

**Fig 4. VCP deficiency disrupts macrophage's proinflammatory response to heat-inactivated *Candida albicans*. (A and B)** Western blot analysis of the phosphorylation of signaling proteins in *Vcp*^fl/fl^ and *Vcp*^fl/fl^ Lyz2-Cre BMDMs stimulated with HKCA-Y (A), ZymD (B), for indicated times. (**C**) *Vcp*^fl/fl^ and *Vcp*^fl/fl^ Lyz2-Cre BMDMs were stimulated with ZymD or HKCA-Y for 2 h, qPCR analysis was performed to quantify the expression of *Tnf-α*, *Il-6*, *Il-12b*, *Il-1b*, *Cxcl1* and *Cxcl2*. (**D**) ELISA analysis of cytokines and chemokines in BMDMs derived from *Vcp*^fl/fl^ and *Vcp*^fl/fl^ Lyz2-Cre mice and left unstimulated (Mock) or stimulated for 24 h with heat-killed *C. albicans* (HKCA) (MOI, 2) or ZymD (100 μg/ml). Data are from one experiment representative of three. Data are shown as mean ± SD of three biological replicates and analyzed by unpaired two-tailed Student's t test (C-D). (*: $p < 0.05$; **: $p < 0.01$; ***: $p < 0.001$; ****: $p < 0.0001$).

BMDMs (**Figs 4B** and **S6B**). These results validated a crucial role of VCP in promoting SYK-dependent signaling upon HKCA treatment.

Next, we sought to investigate how VCP impacts HKCA-induced proinflammatory response. By RT-PCR, we found that the mRNA levels of proinflammatory cytokines (*Il6*, *Tnfa*, *Il1b*, and *Il12b*) and chemokines (*Cxcl1* and *Cxcl2*) were lower in VCP deficient BMDMs compared to those in WT BMDMs after HKCA-Y and ZymD stimulation (**Fig 4C**). Further, the secretion of TNF-α, IL-6 and CXCL-1 measured by ELISA were also reduced in VCP deficient BMDMs stimulated with HKCA-Y and ZymD (**Fig 4D**). Furthermore, the overexpression of VCP did not appear to affect the tyrosine phosphorylation of Dectin-1 (**S7A Fig**). VCP-deficient BMDMs did not exhibit impaired phagocytosis of *C. albican*s (**S7B Fig**), or the translocation of SYK from the cytosol to the cell membrane (**S7C Fig**). We did not observe significant defects in LPS-induced TLR4 or LTA-induced TLR2 responses in VCP-KO

macrophages, suggesting that VCP specifically regulates antifungal immunity (**S7D** and **S7E Fig**). Therefore, VCP is crucially involved in the induction of proinflammatory cytokines and chemokines by HKCA.

## VCP is involved in host defense against *Candia albicans* infection *in vivo*

To assess the function of VCP in antifungal immunity, we intravenously (i.v.) injected $Vcp^{fl/fl}$ and $Vcp^{fl/fl}$ Lyz2-Cre mice with a lethal dose of *C. albicans*. We observed that $Vcp^{fl/fl}$ mice showed moderate weight loss, whereas $Vcp^{fl/fl}$ Lyz2-Cre mice exhibited substantial weight losses after infection with *C. albicans* (**Fig 5A**). Moreover, 90% of the $Vcp^{fl/fl}$ Lyz2-Cre mice

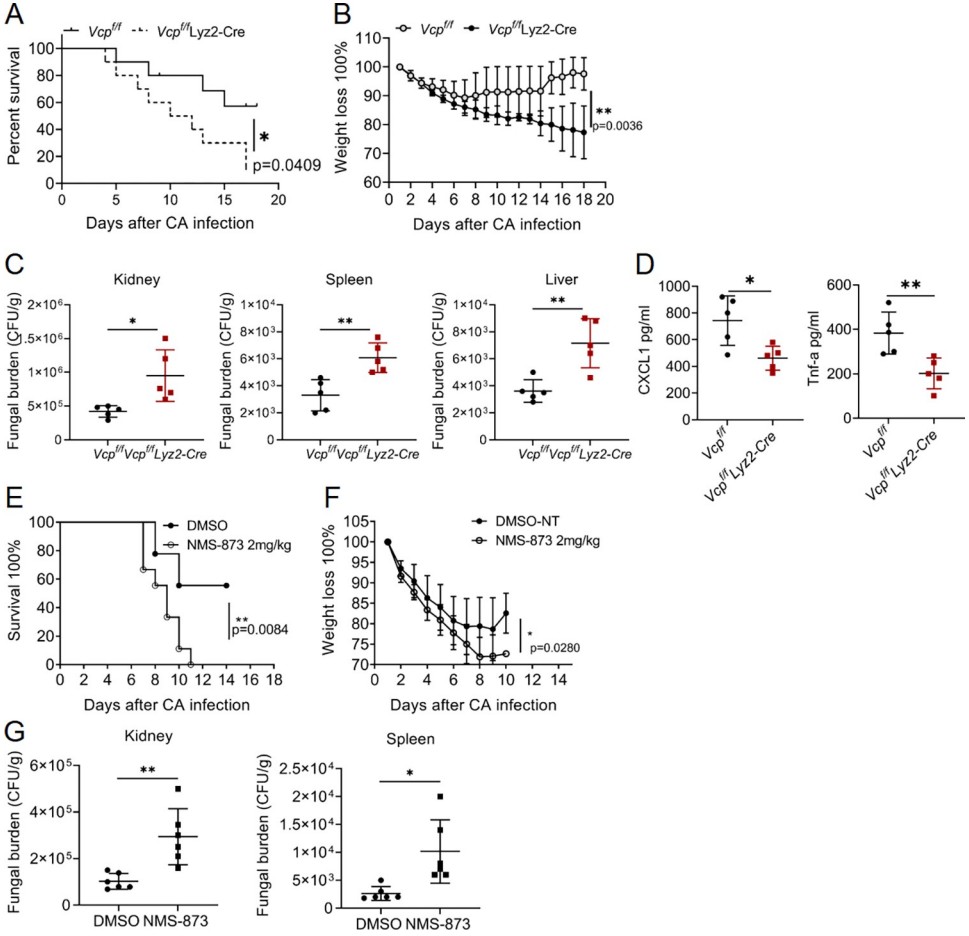

**Fig 5. VCP is involved in host defense against systemic *C. albicans* infections. (A and B)** Survival (**A**) and weight loss (**B**) of $Vcp^{fl/fl}$ and (n = 10) and $Vcp^{fl/fl}$ Lyz2-Cre (n = 10) mice (age-matched littermates) infected with *C. albicans* strain SC5314 ($2 \times 10^5$ fungal cells per mouse) by i.v. injection; weight is presented relative to initial body weight, set as 100%. **(C)** Fungal burdens in the kidneys, livers, and spleens of infected mice 5 d post-infection. The infected organs were homogenized and then serial diluted, and presented as colony-forming units (CFU) per gram of tissue. Dots represent individual mice. **(D)** ELISA measurement of TNF-α and CXCL1 presented in WT and $Vcp^{fl/fl}$ Lyz2-Cre mice sera 24 h post infection with *C. albicans*. Dots represent individual mice. **(E)** 6- to 8-week-old mice (9 pairs of sex- and age-matched littermates) i.p. injected with vehicle (DMSO) or NMS-873 (2 mg/kg) for three consecutive days and then were i.v. infected with *C. albicans* strain SC5314 ($2 \times 10^5$ fungal cells per mouse), and their mortalities (**E**) or weight loss **(F)** were documented. **(G)** Quantification of *C. albicans* in the kidneys and spleens of vehicle or NMS-873 treated mice infected with *C. albicans* strain SC5314 intravenously. Serial dilutions of homogenized tissues were quantified and the data were presented as colony-forming units (CFU) per gram of tissue. $^*p < 0.05$, $^{**}p < 0.01$, $^{***}p < 0.001$, log-rank test was performed (A and E), two-tailed unpaired Student's t test (C-D and G), two-way ANOVA (B and F). Data were pooled from three independent experiments.

died whereas only 40% of WT mice succumbed (**Fig 5B**). Consistently, *Vcp*<sup>fl/fl</sup> Lyz2-Cre mice exhibited more *C. albicans* CFUs in the kidneys, livers, and spleens compared with WT mice after infection with *C. albicans* for 5 days (**Fig 5C**). Moreover, the proinflammatory cytokines TNF-α and chemokine CXCL1 in the serum of *Vcp*<sup>fl/fl</sup> Lyz2-Cre mice were significantly lower than that in WT mice (**Fig 5D**). These results indicate the expression of VCP in the macrophage and neutrophil compartment is critical to mount an anti-Candida immune response for host defense.

Next, we wonder whether the VCP's ATPase activity may be required for anti-fungal defense. After three days consecutive intraperitoneal (i.p.) administration of inhibitor NMS-873 or DMSO control, then mice were intravenously with *C. albicans*. NMS-873 injected mice had greater mortality than DMSO-treated control mice at early stages of infection and suffered more weight loss (**Fig 5E and 5F**). Consistent with those findings, NMS-873 injected mice had more fungal burden in the kidney and spleen (**Fig 5G**). Of note, NMS-873 alone did not exhibit a notable toxicity as mice showing no signs of distress or weight change (**S7F Fig**). These results indicated that inhibition of VCP's ATPase activity can exacerbate the severity of fungal infection. Altogether these data demonstrated that *Vcp* is an important component of antifungal host defense *in vivo*.

## Discussion

The initiation of innate immune signaling pathways is characterized by the assembly of mega signalosomes. In this study, we identified a previously unknown function of VCP/p97 in promoting Dectin-1 mediated signaling events. VCP is vital for cell function and survival [26,29,35], and we demonstrated that VCP also participates in mounting the proinflammatory responses to *C. albicans* infections (**Fig 6**). Biochemical analyses revealed that VCP is recruited to the Dectin1-SYK complex and gets tyrosine phosphorylated by SYK. We also provided compelling evidence suggesting VCP-assisted segregation and degradation of polyubiquitinated phosphatase SHP as novel mechanism regulating Dectin-1 signaling. Further, we found the ATPase activity of VCP is crucial for promoting Dectin-1 signaling. Genetic disruption or chemical blockade of VCP's ATPase activity alleviated SYK activation and the downstream inflammatory responses. Altogether, our work demonstrates ATP-dependent segregase VCP as a novel component of the Dectin-1-SHP2-SYK complex imparting SYK activation and advocating VCP as a potential drug target for antifungal immunity.

Dectin-1 activation induced robust polyubiquitination on SHP-1, and VCP deficiency led to a more prominent interaction between SHP-1 and SYK. SHP-1 functions as the key phosphatase inactivating SYK that is the signaling hub mounting antifungal immunity and thus needs to be tightly controlled. The polyubiquitylated proteins can be direct targets of the proteasome when readily accessible, whereas those embedded into membranes or assembled into complexes, such as the ubiquitinated SHP-1 in the SYK signalosome, need to be extracted or segregated by the conserved VCP with the aid of its cofactor complex, UFD1-NPL4, prior to the proteasomal degradation [36]. We identified the presence of UFD1 in the Dectin1 signalosome, implying an urgent need for UFD1 to detect and pull out those polyubiquitinated proteins. Our results demonstrate that the ubiquitinated SHP-1 seems to be such a substrate of VCP-UFD1 and needs to be processed timely to sustain SYK activation. Our work therefore identified a new mechanism of SYK activity maintenance by controlling the presence of SHP-1 in the Dectin-1/SYK signalosome during antifungal immunity.

The canonical role for VCP as an ATPase is to target and transport the polyubiquitinated substrate proteins to the proteasome for degradation [24,37,38], while the posttranslational modifications regulating VCP activity have not been well understood. Zhu *et al.* shows that

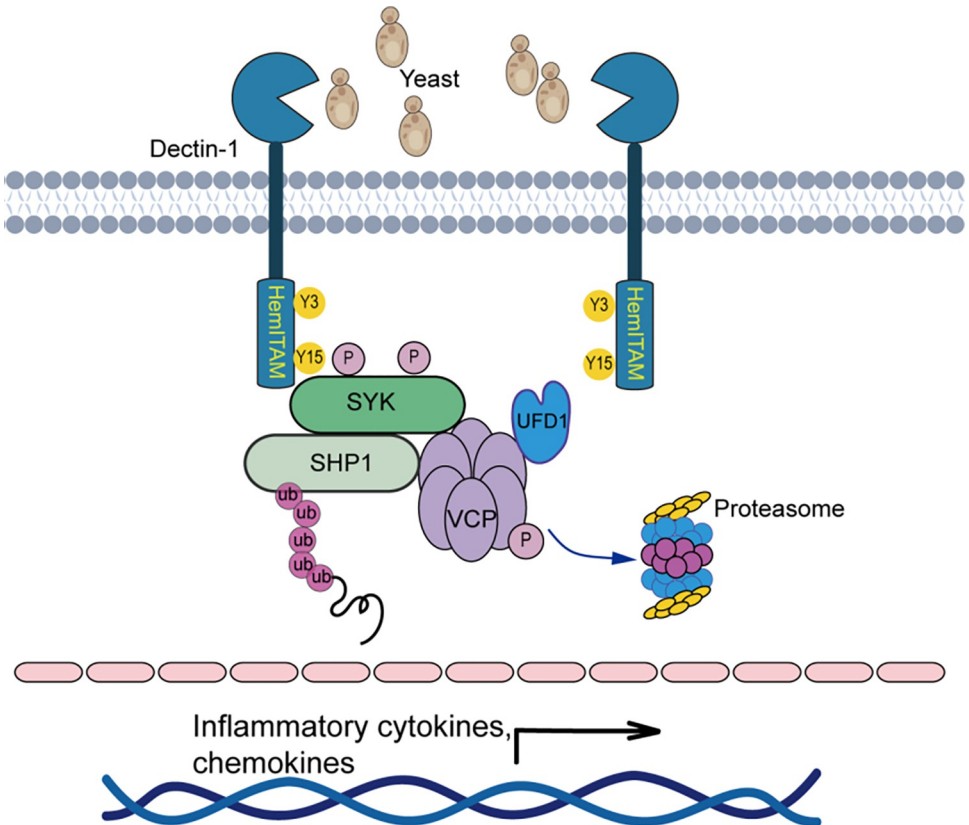

**Fig 6. A work model depicting the mechanism by which VCP promotes SYK-mediated signal activation.** Upon *C. albicans* stimulation, VCP binds to the Dectin-1 signalosome composed of Dectin-1, SYK and SHP-1, where SYK phosphorylates VCP. Subsequently, VCP works together with its partner UFD1 to extract the ubiquitinated phosphatase SHP1 from the signalosome, and this sustains SYK activation.

polo-like kinase 1 (Plk1) as a key mitotic kinase, phosphorylates Thr76 in VCP, and this phosphorylation recruits VCP to the centrosome to regulate its orientation [39]. Wang *et al*. demonstrates that autophagy-inducing kinases ULK1 and ULK2 phosphorylate VCP, therefore promoting VCP activity to disassemble stress granules, a process that is essential for controlling inclusion body myopathy (IBM) progress [29]. In our work, we found that Dectin-1 activation induced the interaction of SYK and VCP, and this binding promoted tyrosine phosphorylation of VCP by SYK. It remains to be determined whether SYK-induced VCP phosphorylation might be involved in initiating its ATPase activity, and hence able to segregate the polyubiquitinated SHP1 from the signalosome. Less SHP1 presence subsequently enhances SYK activation, while this hypothesis awaits further experimental affirmation.

In summary, we identified a previously unknown role for VCP in the activation of SYK and antifungal immune responses and propose VCP as a potential drug target in treating fungal infections.

## Materials and methods

### Ethics statement

All the mice were bred and maintained in a specific-pathogen-free animal facility at Shanghai Institute of Immunity and Infection. All the procedures were conducted in compliance with a

protocol approved by the Institutional Animal Care and Use Committee at Shanghai Institute of Immunity and Infection.

## Mice

*Vcp*<sup>fl/fl</sup> mice (C57BL/6 background) generated by Crispr-Cas9 (Cyagen) were bred onto Lyz2-Cre mouse strain (from the Jackson Lab) to generate *Vcp*<sup>fl/fl</sup> Lyz2-Cre mice in this study. The *Vcp* floxed mice were generated by Cyagen Biosciences (Suzhou, China), with the exons 4~5 flanked with two LoxP alleles. Simply, construct a Cas9-sgRNA expression plasmid and a targeting vector containing a loxP site. Then, insert the plasmid upstream and the targeting vector downstream of exons 4–5. Microinject the linearized Cas9-sgRNA and targeting vector into mouse zygotes and implant the injected zygotes into C57BL/6 female mice. Genotyping was performed by PCR using the following primers: forward 5'-CCTAGTGGGGAGGAGCA TAGTTT-3', reverse 5'- GATTTCAGAGACTGAATAGTCCTGGC –3'. Lyz2-Cre mice were obtained from The Jackson Laboratory. The primers for Lyz2-Cre transgenic mice genotyping were oIMR3066 mutant 5'-CCCAGAAATGCCAGATTACG-3', oIMR3067 common 5'-CTTGGGCTGCCAGAATTTCTC-3', oIMR3068 Wild type 5'-TTACAGTCGGCCAGGCT GAC-3'.

## Cells

Human HEK293T and HEK293 cells (all from American Type Culture Collection) were cultured in DMEM (Gibco, Cat# C11995500BT) plus 10% fetal bovine serum (FBS, Gibco, Cat# 10099141), 100 U/mL penicillin, and 100 mg/mL streptomycin. All the cell lines were regularly checked for mycoplasma contamination by commercial PCR (Lonza Mycoalert, Cat# LT07-418).

Bone marrow-derived dendritic cells (BMDCs) were differentiated from bone marrow cells from 8–10 weeks mice as described previously [40]. Red blood cells were lysed with ACK lysis buffer (0.15 M NH4Cl, 1 mM KHClO3 and 0.1 mM Na2EDTA, pH 7.3). Bone marrow cells were cultured in RPMI-1640 medium containing 10% FBS (HyClone), 2 mM L-glutamine and 200 μM β-mercaptoethanol, IL-4 (10 ng/ml; Biolegend) and GM-CSF (20 ng/ml; Biolegend) for 9 days. The medium containing GM-CSF and IL-4 was replenished every 2 days. On day 9, nonadherent cells were collected by centrifugation and then resuspended in fresh medium containing GM-CSF and IL-4 for use.

Bone marrow-derived macrophages (BMDMs) were differentiated from bone marrow cells from 8–10 weeks mice. Briefly, bone marrow cells were flushed from the femurs and tibias with PBS and dispersed by repeated pipetting. Red blood cells were lysed with ACK lysis buffer (0.15 M $NH_4Cl$, 1 mM $KHClO_3$ and 0.1 mM $Na_2EDTA$, pH 7.3). Bone marrow cells were cultured in RPMI-1640 medium supplemented with 30% L929-conditioned medium (containing the cytokine M-CSF), as well as 10% FBS. At day 4, nonadherent cells were removed and the fresh RPMI medium with L929-conditioned medium was added. BMDMs were used on day 7.

## Regents and Antibodies

Zymosan, Zymosan Deplete, Pam3CSK4 and CpGB were purchased from InvivoGen; LPS, poly(I:C) were purchased from sigma; NMS-873 (S7285), PRT (S7738) were purchased from Selleck. PP2 (HY-13805), CB-5083 (HY-12861), DBeQ (HY-15945), ML240 (HY-19795) and TPI-1 (HY-100463) were purchased from MCE. Mouse anti-V5 (66007-I-lg) was from Proteintech; mouse anti-c-Myc (TA150121-1) was from OriGene; rabbit anti-c-Myc (A190-105A) was from Bethl Laboratories; rabbit anti-Flag (PA1-984B) was from Sigma-Aldrich; mouse anti-FLAG (M2) was from Sigma-Aldrich; rabbit anti-HA was from Rockland

Immunochemicals; rabbit anti-VCP (2648), anti–p-Syk (Tyr525/526) (2710), anti-Syk (13198), anti-phospho-Tyrosine (9411), anti-p-PLC-γ2 (Tyr759) (3874), anti-PLC-γ2 (3872), anti-p-PKCδ (Thr505) (9374), anti-PKCδ (9616), anti-IκBα (44D4) (4812), anti-p-IκBα (Ser32) (2859), anti-p-NF-κB p65 (Ser536) (3303), anti-NF-κB p65 (8242), anti-p-p38 MAPK (Thr180/Tyr182) (9211), anti-p38 MAPK (8690), anti-p-SAPK/JNK (Thr183/Tyr185) (4668), anti-SAPK/JNK (9252), anti-p-Erk1/2 (Thr202/Thr204) (4370), anti-Erk1/2 (4695), were from Cell Signaling Technology; PTPN6/SHP1 polyclonal antibody (24546-1-AP) was from Proteintech: horseradish peroxidase-conjugated secondary Abs and anti-β-actin were from Jackson Laboratory; anti-SHP-2 (sc-7384), anti-VCP (sc-57492), anti-UFD1 (sc-377265), anti-SHIP1 (sc-8425), anti-IκBα (sc-371) and Protein A/G PLUS-Agarose (sc-2003) were from Santa Cruz Biotechnology.

## Plasmid construction and transfection

The cDNA encoding VCP was amplified from C57BL/6 bone marrow cells by primers 5′-GCCTCTGGAGCCGATTC-3′ (F) and 5′-TTAGCCATACAGGTCATCGTCAT-3′ (R), and was cloned into the pCDNA3.1 vector through the BamHI and NotI sites. The pCDH-Flag-VCP was generated by subcloning the VCP coding-sequence into the pCDH-IRES-Puro vector. Plasmids V5-SYK and HA-dectin-1 were cloned into the pCDNA3.1 vector as described [9]. All constructs were confirmed by DNA sequencing. Other plasmids used in this study were as described previously. For transient transfection of plasmids into HEK293T cells, lipofectamine 2000 reagents (Invitrogen) were used. Seed HEK293T cells to be 70% confluent in 6 cm dish. Dilute amounts of Lipofectamine Reagent in Opti-MEM Medium and meanwhile dilute 3ug DNA in Opti-MEM Medium per 6cm dish. Add diluted DNA to diluted Lipofectamine 2000 Reagent (1:1 ratio) and then incubate for 15 minutes. Lastly, add DNA-lipid complex to cells.

## RNA interference assay and RNA quantitative real-time PCR

The siRNA sequences are shown in S2 Table. These siRNA duplexes were transfected into mouse BMDMs cells using Lipofectamine RNAiMAX Transfection Reagent (Invitrogen, 13778150) according to the manufacturer's instructions. siRNA (final concentration, 100 nM) reagents were transfected into each well.

Total RNA from cells was extracted using the RNA Fast 200 Extraction Kit, according to the manufacturer's instructions (Fastagen, Shanghai, China). Total RNAs from various tissues were extracted with TRIzol reagent according to the manufacturer's instructions (Invitrogen). cDNA was reverse transcribed from 500ng of total RNA with a PrimeScript RT-PCR Kit (Takara Bio, Cat#RR014). RT-PCR was carried out with the primer pairs listed below on an Applied Biosystems 7900HT Fast Real-Time PCR System. RT-PCR was performed with SYBR Green Real-time PCR Master Mix (Yeasen, Cat# 11202ES08), The cycling conditions were denaturation at 95°C for 3 min, followed by 40 cycles of standard PCR. The specificity of the amplified products was determined by melting curve analysis. The $2^{-\Delta\Delta Ct}$ method was used to calculate relative expression changes. Gene expression values were normalized against GAPDH or β-*actin* mRNA expression. GAPDH or β-*actin* housekeeping gene in each sample. The primers are listed in the Supporting Information S1 Table.

## Lentivirus preparation and infection

The lentiviral vector pCDH-Puro expressing VCP or empty were transiently transfected into HEK293T cells along with packaging plasmids (Δ8.91/VSV-G) at a ratio of pCDH:Δ8.91: VSVG = 4:3:2, and 48 hours later virus-containing medium was harvested and passed through

a 0.45 μm filter. On day 2 of BMDM differentiation, lentiviruses were added to the BMDMs and incubated for 12 hours. Subsequently, virus-containing medium was removed and replaced with fresh complete medium. In 48 h, lentivirus-infected BMDMs were selected under 2 μg/ml puromycin for 7 days. The selected BMDMs were re-plated and subjected to stimulation the next day.

## ELISA

BMDCs (differentiated with GM-CSF plus IL-4) were stimulated for 24 h with ZymD or ZymA. BMDMs were primed overnight with either GM-CSF (10 ng/ml) or IL-4 (10 ng/ml), followed by stimulation for 24 h with ZymD, Curdlan or heat-killed *C. albicans*, respectively. Supernatants were harvested and the amounts of TNF, IL-6, CXCL1 secreted were measured by ELISA kits according to the manufacturers' instructions (eBioscience). Following intravenous infection with $2 \times 10^5$ cells of live *C. albicans* strain SC5314 for 24 h, mouse serum was collected, the levels of cytokines and chemokines were measured by ELISA.

## Co-immunoprecipitation and immunoblot analysis

Cells were lysed in lysis buffer (50 mM Tris-HCl [pH 7.4], 150 mM NaCl, 1% Triton X-100, and 1 mM EDTA [pH 8.0]) supplemented with protease inhibitor cOmplete mini (Roche) and 1 mM PMSF, 1 mM Na3VO4, and 1 mM NaF for 30 minutes on ice, and cell debris was removed by centrifugation at 13,000 rpm for 15 minutes. The supernatants were collected and incubated with protein A/G Plus-Agarose and 2 μg of immunoprecipitation antibodies. After incubation at 4°C for 12 h, the beads were washed with 500 μl of lysis buffer four times. Immunoprecipitated proteins were eluted by boiling in 1% (w/v) SDS sample buffer. For Western blot analysis, immunoprecipitates or whole-cell lysates were resolved by 10% SDS-PAGE, and transferred onto PVDF membranes for immunoblotting with the indicated antibodies.

## Immunofluorescence

Cells were fixed with 4% paraformaldehyde and followed by permeabilization with PBS containing 0.1% Triton X-100 for 10 minutes. Next, the cells were incubated with 10% goat serum at room temperature for 1 hour prior to 12-h incubation with primary antibody at 4°C. Afterwards, the probed samples were washed three times with ice cold PBS and then stained with fluorophore (Alex 488 or Alex 405 and Alexa 594 conjugated secondary antibodies). After staining, the samples were counter-stained with DAPI and sealed on a slide with nail polish. Sealed slides were analyzed using OLYMPUS FV3000 microscope with companion software.

## BMDMs phagocytosis assay

For GFP-C. albicans, fungi were fixed with 2% paraformaldehyde at room temperature for 30 mins, then co-cultured with BMDMs for the various times as indicated in the figure legend. Unbound yeasts were gently washed 5 times with 1×PBS, and then analyzed by flow cytometry.

## *C. albicans* infection

*C. albicans* SC5314 were cultured in YPD broth overnight at 30°C on a shaking platform, washed, and resuspended in PBS and then counted using a LUNA automated fluorescence cell counter. Live *C. albicans* cells ($2 \times 10^5$ yeast cells in 0.1 ml of PBS buffer) were injected intravenously into 6- to 8-week-old mice. Infected mice were monitored daily for weight loss and survival. Fungal burdens were measured 5 days after infection. Specifically, the kidneys, spleens and livers were collected and homogenized. The tissue homogenates were serially diluted and

plated on yeast extract–peptone–dextrose agar. Fungal colony-forming units were counted after 24 hours. Heat-killed (HK) *C. albicans* were generated by incubating *C. albicans* yeast or hyphae cells in PBS in a heat block at 95˚C for 10 min.

## Statistical analysis

The statistical significance between two groups was determined by unpaired two-tailed Student's t test; multiple-group comparisons were performed using one-way ANOVA; and the weight changes were analyzed by two-way ANOVA for multiple comparisons. $P < 0.05$ was considered to be significant. The results are shown as mean, and the error bar represents SD of mean biological or technical replicates as indicated in the figure legend. The p values represented as $^*p < 0.05$, $^{**}p < 0.01$, $^{***}p < 0.001$, $^{****}p < 0.0001$. We performed the statistical analyses by using GraphPad Prism 8.

## Supporting information

**S1 Fig. VCP is tyrosine phosphorylated by Dectin-1 signaling reconstituted in HEK293T cells.** (A) HEK293T cells were transfected with the plasmids expressing Dectin-1, SYK, empty control or VCP plasmids for 36 h and then stimulated with ZymD (100 μg/ml) for indicated time. Cell lysates were subjected to IP with anti-pTyr and then immunoblotted with indicated antibodies. (B) HEK293T cells were transfected with the plasmids expressing Dectin-1, SYK, empty control or VCP plasmids for 36 h and then stimulated with ZymD (100 μg/ml) for indicated time. Cell lysates were subjected to IP with anti-Flag and then immunoblotted with indicated antibodies. In A-B, one representative experiment of three independent experiments is shown.
(TIF)

**S2 Fig. Quantification of VCP regulated p-SYK and p-ERK in Dectin-1 signaling.** (A) Densitometric quantification of p-Syk, p-PLCγ2, p-Erk, and p-p38 from three independent experiments, as depicted in Fig 1E, was conducted using ImageJ. (B) Densitometric quantification of p-Syk, p-PLCγ2, p-PKCδ, p-Jnk, p-Erk, p-p38 and p-IKKα/β from three independent experiments, as shown in Fig 2B, was performed using ImageJ. Data are shown as mean ± SD and were analyzed by one-way ANOVA (A and B). (*: p<0.05; **: p<0.01; ***: p<0.001; ****: p<0.0001, ns: no significance).
(TIF)

**S3 Fig. Inhibition of VCP's ATPase dampens the activation of antifungal signaling pathway.** (A) Bone marrow-derived macrophages (BMDMs) from wild-type mice were pretreated with DMSO, NMS-873 (2 μM), or DBeQ (10 μM) for 1 hour, and then stimulated with Zymosan D (ZymD) (100 μg/ml) for the indicated times. This was followed by western blot analysis of the indicated proteins. (B) ELISA was performed for TNF-α, IL-6, and CXCL1 in BMDMs derived from wild-type mice, which were left unstimulated (Mock) or stimulated for 24 hours with ZymD (100 μg/ml) or Zymosan A (ZymA) (100 μg/ml), with DMSO or 2 μM of DBeQ. *: p<0.05; **: p<0.01; ***: p<0.001; ****: p<0.0001 based on two-tailed unpaired Student's t test (B).
(TIF)

**S4 Fig. VCP targets SHP-1 for proteosome-dependent degradation.** (A) Co-immunoprecipitation (Co-IP) analysis of the interaction between VCP and SHP1 in HEK293T cells transfected with Flag-VCP and V5-SHP1. (B) HEK293T cells were transfected with plasmids expressing SHP1, SYK, an empty control, or Dectin-1 for 36 hours and then stimulated with ZymD (100 μg/ml) for the indicated time. Cell lysates were immunoblotted with the indicated

antibodies. (C) V5-SHP1 was transfected into HEK293T cells together with Flag-VCP and then treated with the protein synthesis inhibitor CHX for the indicated times. The protein levels of SHP1 were detected by immunoblot analysis. (D) V5-SHP1 was transfected into HEK293T cells together with Flag-VCP, followed by treatment with MG132 (10 μM) or chloroquine (CQ) (50 μM) for 10 hours. The protein levels of SHP1 were detected by immunoblot analysis. One representative experiment of three independent experiments is shown.
(TIF)

**S5 Fig. VCP promotes the dissociation of SHP-1 from the SYK signalosome.** (A) Wild-type (WT) and VCP-deficient bone marrow-derived macrophages (BMDMs) were stimulated with ZymD (100 μg/ml) for the indicated times, and the cell lysates were subjected to immunoprecipitation (IP) with anti-SHP1 antibody (Ab), followed by Western blot analysis with anti-Syk Ab. (B) HEK293T cells were transfected with Myc-SYK, V5-SHP1, and Flag-VCP. They were then subjected to IP with anti-V5, and probed with the indicated antibodies (left margins). (C) WT and VCP-deficient BMDMs were transfected with si-NC or si-SHP1 for 5 days and then stimulated with ZymD for the indicated time points, followed by Western blot analysis of the indicated proteins. (D) WT and VCP-deficient BMDMs were pretreated with DMSO or the SHP1 inhibitor TPI-1 (10 μM) for 1 hour and then stimulated with ZymD for the indicated time points, followed by Western blot analysis of the indicated proteins. Data are representative of three independent experiments.
(TIF)

**S6 Fig. Quantification of p-SYK in WT and VCP-deficient macrophages.** (A, B) Densitometric quantification of p-Syk, p-PLCγ2, p-PKCδ, p-JNK, and p-Erk, as shown in Fig 4A and 4B, was measured using ImageJ. The data are presented as mean ± SD and were analyzed using an unpaired two-tailed Student's t-test. (A-B). (*: $p < 0.05$; **: $p < 0.01$; ***: $p < 0.001$; ****: $p < 0.0001$, ns: no significance)
(TIF)

**S7 Fig. Phagocytosis and TLR2/4 responses are normal in VCP-deficient macrophages.** (A) HEK293T cells were transfected with plasmids expressing Dectin-1, SYK, an empty control, or VCP for 36 hours and then stimulated with ZymD (100 μg/ml) for the indicated times. Cell lysates were subjected to immunoprecipitation (IP) with anti-HA and then immunoblotted with the indicated antibodies. (B) Phagocytosis of $Vcp^{fl/fl}$ or $Vcp^{fl/fl}$ Lyz2-Cre BMDMs was evaluated by the method described in the Materials and Methods section. (C) $Vcp^{fl/fl}$ or $Vcp^{fl/fl}$ Lyz2-Cre cells were treated with HKCA (MOI = 1) for the indicated times, followed by immunofluorescence staining for the indicated proteins. Scale bar = 5 μm. (D-E) BMDMs from wild-type (WT) control mice or Vcp-deficient mice were stimulated with LPS (200 ng/ml) or LTA (100 μg/ml) for the indicated times, followed by Western blot analysis of the indicated proteins. (F) Six- to eight-week-old mice (5 pairs of sex- and age-matched littermates) were intraperitoneally (i.p.) injected with vehicle (DMSO) or NMS-873 (2 mg/kg) for three consecutive days, and their weight change was documented. Data are presented as mean ± SD and were analyzed using an unpaired two-tailed Student's t-test (B), and two-way ANOVA (F) *: $p < 0.05$; **: $p < 0.01$; ***: $p < 0.001$, ns: no significance.
(TIF)

**S1 Table. qPCR primers used in this study. (DOCX format).**
(DOCX)

**S2 Table. siRNA oligos used in this study. (DOCX format).**
(DOCX)

**S3 Table. The entire MS data, including protein and peptide information from primary BMDC cells, related to Fig 1. (XLSX format).**
(XLSX)

**S1 Data. Western blot raw data.**
(ZIP)

## Acknowledgments

We would like to express our gratitude to all the staff members of the institutional core facilities and technical platforms, particularly the animal facility management and husbandry staff at the Shanghai Institute of Immunity and Infection. We also extend our thanks to the Translational Medicine Core Facility of Shandong University for their consultation and for providing access to essential instruments that supported this work. Additionally, we are sincerely grateful to all the lab members in Drs. Xiao and Gao's laboratories for their helpful discussions and suggestions.

## Author Contributions

**Conceptualization:** Tian Chen, Chengjiang Gao.

**Data curation:** Zhugui Shao.

**Formal analysis:** Zhugui Shao, Li Wang, Limin Cao, Tian Chen, Xin-Ming Jia, Wanwei Sun, Chengjiang Gao, Hui Xiao.

**Funding acquisition:** Hui Xiao.

**Investigation:** Zhugui Shao, Chengjiang Gao, Hui Xiao.

**Methodology:** Zhugui Shao, Wanwei Sun.

**Project administration:** Zhugui Shao, Li Wang, Limin Cao, Tian Chen, Xin-Ming Jia, Wanwei Sun, Chengjiang Gao, Hui Xiao.

**Resources:** Zhugui Shao, Li Wang, Limin Cao, Tian Chen, Xin-Ming Jia, Wanwei Sun, Chengjiang Gao, Hui Xiao.

**Software:** Zhugui Shao, Wanwei Sun.

**Supervision:** Chengjiang Gao, Hui Xiao.

**Validation:** Zhugui Shao.

**Visualization:** Zhugui Shao.

**Writing – original draft:** Zhugui Shao, Tian Chen, Chengjiang Gao, Hui Xiao.

**Writing – review & editing:** Zhugui Shao, Wanwei Sun, Chengjiang Gao, Hui Xiao.

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
