## [Decision Letter · Decision Letter 0]

9 Sep 2024

Dear Dr. Gao,

Thank you very much for submitting your manuscript "The protein segregase VCP/p97 promotes host antifungal defense via regulation of SYK activation" for consideration at PLOS Pathogens. As with all papers reviewed by the journal, your manuscript was reviewed by members of the editorial board and by several independent reviewers. The reviewers appreciated the attention to an important topic. Based on the reviews, we are likely to accept this manuscript for publication, providing that you modify the manuscript according to the review recommendations.

The authors addressed most of previous reviewers' editorial and experiment comments in this revised manuscript. The reviewers also agreed that the revised manuscript is much improved. Having said that, they still raised some concerns that need to be addressed in the second round of revision, as can be seen in the reviewers' comments. These include more quantitative measurement of VCP phosphorylation (reviewer 1), providing raw blotting data for Ig pulldown (reviewer 3), and several comments from reviewer 2. I think that the authors could reasonably address these questions.

Sincerely,

Yong-Sun Bahn, Ph.D.

Guest Editor

PLOS Pathogens

Michal Olszewski

Section Editor

PLOS Pathogens

Michael Malim

Editor-in-Chief

PLOS Pathogens

orcid.org/0000-0002-7699-2064

The authors addressed most of previous reviewers' editorial and experiment comments in this revised manuscript. The reviewers also agreed that the revised manuscript is much improved. Having said that, they still raised some concerns that need to be addressed in the second round of revision, as can be seen in the reviewers' comments. These include more quantitative measurement of VCP phosphorylation (reviewer 1), providing raw blotting data for Ig pulldown (reviewer 3), and several comments from reviewer 2. I think that the authors could reasonably address these questions.

Reviewer Comments (if any, and for reference):

Reviewer's Responses to Questions

**Part I - Summary**

Reviewer #1: This is a revised manuscript and the authors responded to reviewers' point by providing additional experiments and adding more robust statistic analysis. Overall responses are acceptable. However, the phosphorylation of VCP upon dectin-1 signaling is not really convincing based on western blot experiment in fig 1. What's difference between no activation (0 zyme) vs activation (Zymo treatment)?

They need to provide clear, convincing evidence to make the case.

Reviewer #2: The revised submission has addressed most of my concerns and the quality has been improved. Impressive amount of work has been included. I have a few additional comments.

1. It is nice to have the ATPase activity-dead VCP mutant. Not sure whether such mouse mutant strain is available. To confirm the ATPase activity of VCP is required for fungal infectivity, in addition to using the NMS-872 inhibitor, testing the VCP (K542A) ATPase-dead mutant mouse strain would be more convincing. What is the rationale to use NMS-872 at 2 mg/Kg dosage?

2. Line 401, please cite a reference on the role of VCP as an ATPase in targeting and transporting ubiquitinated substrate for degradation.

3. Figure 2B. The image used for the ikBa signals is partially cut out and should use a better image with complete bands. Also comparing with the original blot (have two strong bands), the up band was cut out. Please explain what is the up band and why was it cut out?

4. Figure 3E. please add the label to the right panel.

5. Figure 6. In the model, the letters on the dectin-1 phosphorylation site are hard to see. May change the color of the letters.

Reviewer #3: The authors’ revisions, including toning down their language regarding the role of VCP in regulation of Dectin-1 signaling, quantification of experimental replicates, and new data have strongly improved the manuscript. This is an important study identifying a new regulator of CLR signaling. The authors have addressed our concerns well and their data now fit their conclusions.

**Part II – Major Issues: Key Experiments Required for Acceptance**

Reviewer #1: Further verification of phosphorylaion of VCP upon dectin-1 activation is needed.

Reviewer #2: (No Response)

Reviewer #3: (No Response)

**Part III – Minor Issues: Editorial and Data Presentation Modifications**

Reviewer #1: (No Response)

Reviewer #2: (No Response)

Reviewer #3: (No Response)

PLOS authors have the option to publish the peer review history of their article (what does this mean?). If published, this will include your full peer review and any attached files.

Reviewer #1: No

Reviewer #2: No

Reviewer #3: No

Figure Files:

Data Requirements:

Reproducibility:

References:

---

## [Decision Letter · Decision Letter 1]

17 Oct 2024

Dear Dr. Gao,

We are pleased to inform you that your manuscript ' The protein segregase VCP/p97 promotes host antifungal defense via regulation of SYK activation ' has been provisionally accepted for publication in PLOS Pathogens.

Best regards,

Yong-Sun Bahn, Ph.D.

Guest Editor

PLOS Pathogens

Michal Olszewski

Section Editor

PLOS Pathogens

Michael Malim

Editor-in-Chief

PLOS Pathogens

orcid.org/0000-0002-7699-2064

The two original reviewers appreciated that the authors address their concerns and comments nicely.

Reviewer Comments (if any, and for reference):

Reviewer's Responses to Questions

**Part I - Summary**

Reviewer #1: The authors addressed the issues raised by the reviewers well.

Reviewer #2: The authors have addressed my concerns. This is a nice study with impressive amount of data.

**Part II – Major Issues: Key Experiments Required for Acceptance**

Reviewer #1: (No Response)

Reviewer #2: (No Response)

**Part III – Minor Issues: Editorial and Data Presentation Modifications**

Reviewer #1: (No Response)

Reviewer #2: (No Response)

PLOS authors have the option to publish the peer review history of their article (what does this mean?). If published, this will include your full peer review and any attached files.

Reviewer #1: No

Reviewer #2: No

---

## [Editor Report · Acceptance letter]

24 Oct 2024

Dear Dr. Gao,

We are delighted to inform you that your manuscript, " The protein segregase VCP/p97 promotes host antifungal defense via regulation of SYK activation ," has been formally accepted for publication in PLOS Pathogens.

Best regards,

Michael Malim

Editor-in-Chief

PLOS Pathogens

orcid.org/0000-0002-7699-2064